# Corneal Sub-Basal Nerve Plexus in Non-Diabetic Small Fiber Polyneuropathies and the Diagnostic Role of In Vivo Corneal Confocal Microscopy

**DOI:** 10.3390/jcm12020664

**Published:** 2023-01-13

**Authors:** Anna M. Roszkowska, Adam Wylęgała, Ludovica Gargiulo, Leandro Inferrera, Massimo Russo, Rita Mencucci, Bogusława Orzechowska-Wylęgała, Emanuela Aragona, Maura Mancini, Angelo Quartarone

**Affiliations:** 1Ophthalmology Clinic, Department of Biomedical Sciences, University of Messina, 98182 Messina, Italy; 2Ophthalmology Department, Faculty of Medicine and Health Sciences, Andrzej Frycz Modrzewski Krakow University, 31-327 Krakow, Poland; 3Adam Wylęgała, Health Promotion and Obesity Management, Pathophysiology Department, Medical University of Silesia, 40-055 Katowice, Poland; 4Eye Clinic, Department of Medical, Surgical Sciences and Health, University of Trieste, 34100 Trieste, Italy; 5Neurology Clinic, Department of Biomedical Sciences, University of Messina, 98182 Messina, Italy; 6Eye Clinic, Department of Neuroscience, Psychology, Pharmacology and Child Health (NEUROFARBA), University of Florence, 50139 Florence, Italy; 7Clinic of Otolaryngology, Head and Neck Surgery, Department of Pediatric Surgery, Medical University of Silesia, 40-055 Katowice, Poland; 8Ophthalmology Clinic, IRCCS San Raffaele Scientific Institute, Vita Salute San Raffaele University, 20132 Milan, Italy; 9IRCCS Neurological Center Bonino-Pulejo, 98182 Messina, Italy

**Keywords:** small fibers polyneuropathies, corneal subbasal nerve plexus, IVCM, corneal nerves parameters

## Abstract

In vivo corneal confocal microscopy (IVCM) allows the immediate analysis of the corneal nerve quantity and morphology. This method became, an indispensable tool for the tropism examination, as it evaluates the small fiber plexus in the cornea. The IVCM provides us with direct information on the health of the sub-basal nerve plexus and indirectly on the peripheral nerve status. It is an important tool used to investigate peripheral polyneuropathies. Small-fiber neuropathy (SFN) is a group of neurological disorders characterized by neuropathic pain symptoms and autonomic complaints due to the selective involvement of thinly myelinated Aδ-fibers and unmyelinated C-fibers. Accurate diagnosis of SFN is important as it provides a basis for etiological work-up and treatment decisions. The diagnosis of SFN is sometimes challenging as the clinical picture can be difficult to interpret and standard electromyography is normal. In cases of suspected SFN, measurement of intraepidermal nerve fiber density through a skin biopsy and/or analysis of quantitative sensory testing can enable diagnosis. The purpose of the present review is to summarize the current knowledge about corneal nerves in different SFN. Specifically, we explore the correlation between nerve density and morphology and type of SFN, disease duration, and follow-up. We will discuss the relationship between cataracts and refractive surgery and iatrogenic dry eye disease. Furthermore, these new paradigms in SFN present an opportunity for neurologists and clinical specialists in the diagnosis and monitoring the peripheral small fiber polyneuropathies.

## 1. Introduction

Small fibers neuropathies (SFN) are neurological disorders with a significant impact on the patient’s health status and quality of life. The cornea is innervated by small fibers originating from the trigeminal nerve branch that forms the sub-basal nerve plexus (SBNP), so the SFN might affect corneal nerves with alteration of the ocular surface homeostasis and health. Efficient corneal innervation is mandatory for corneal health and ocular surface homeostasis, and it can be investigated by in vivo corneal confocal microscopy (IVCM). This examination permits the immediate analysis of the corneal nerves density and morphology and is an indispensable tool for the corneal structure examination. As it evaluates the SBNP in the cornea, the IVCM provides information about the peripheral small fibers nerves status elsewhere, and it becomes an important tool to investigate peripheral SFN [1,2,3,4,5].

Small-fiber neuropathy is a clinical syndrome characterized by neuropathic pain symptoms and autonomic complaints due to the selective involvement of thinly myelinated Aδ-fibers and unmyelinated C-fibers. Many etiologies and very likely more than one pathogenesis may cause SFN, therefore an accurate diagnosis of SFN is important as it provides a basis for etiological work-up and treatment decisions. The most common causes of SFN are diabetes mellitus, HIV, metabolic syndrome, amyloidosis, Fabry disease, coeliac disease, sarcoidosis, and other systemic illnesses such as hypothyroidism. SFN remains idiopathic in a substantial proportion of patients. The diagnosis of SFN is sometimes challenging as the clinical picture can be difficult to interpret and standard electromyography is normal. In cases of suspected SFN, measurement of intraepidermal nerve fiber density through a skin biopsy and/or analysis of quantitative sensory testing can enable diagnosis. On the other hand, new diagnostic techniques (including measurement of nerve fiber density using corneal confocal microscopy, and nociceptive evoked potentials) may contribute to the diagnostic work-up.

The purpose of the present review is to review the current knowledge regarding corneal nerves in different SFNs in an attempt to assess the correlation between the nerve density and morphology in different types of SFN and disease duration and to highlight the role of IVCM in SFN evaluation and monitoring. We considered reports with confocal studies of SBNP in small fiber polyneuropathies. We excluded the literature related to IVCM in diabetes and its role in diabetic polyneuropathy as it was extensively studied, and the results were summarized [1,2,3,4].

We believe that a complete review would improve our knowledge regarding the corneal health status in SFN and that reviewing all of the available data in the literature could be useful in the diagnosis and monitoring of peripheral small fiber polyneuropathies. Additionally, this information could be useful for ophthalmologists when planning cataract and refractive surgery procedures that may induce or aggravate dry eye disease or induce neurotrophic keratitis with poor visual outcomes.

## 2. Materials and Methods

### 2.1. Small Fibres Neuropathy

Small fibres neuropathy (SFN) is defined as a structural abnormality of small fibres characterized pathologically by degeneration of the distal terminations in myelinated (Aδ) and unmyelinated (C) nerve fibres of the peripheral nerves [6].

Nowadays SFN is considered a distinctive disease. An epidemiological study conducted in a region in the southern part of the Netherlands between 2006 and 2011, showed a prevalence of 53 per 100,000 population with an incidence of 11.7 in the years 2010 and 2011 [7].

Physiologically, the small fibres play an important role in the functionality of the autonomic nervous system as they mediate cardiovascular, gastrointestinal, urogenital, sudomotor, thermoregulatory, and other autonomic functions. In addition, the small-caliber fibres are responsible for transmitting information relating to pain, itching, and temperature [8].

Similarly to other peripheral nerve diseases, the presentation patterns of small fibres neuropathies can be distinguished in the classic length-dependent pattern or non-length-dependent forms. Typically, in the first case, the symptoms begin in the most distal parts of the lower limbs and then slowly rise, progressively affecting the most proximal parts and the upper limbs [9,10].

In non-length-dependent SFN the nerves can be involved individually or according to a multi-neuropathic distribution pattern. The mouth, scalp, face, tongue, or different parts of the trunk are frequently affected [11]. Some rare conditions such as insensitivity to pain and anhidrosis are characterized by the lack of innervation of the skin (C and A-delta fibres) resulting in a lack of sensitivity to pain or sweating spread throughout the body [12].

Symptoms reported by the patient are more often positive sensory symptoms such as burning, aching, electric-like, prickling, or itching sensations. Sometimes, patients complain of negative symptoms, for example, absent or reduced pain and temperature sensation [13].

When the disease involves the autonomic nervous system fibres, the clinical picture becomes more complex and symptoms such as diarrhoea, constipation, micturition problems (e.g., incontinence and hesitation) dry eyes, dry mouth, dizziness on standing from sitting or supine position, palpitations or changes in sweating may be present [14].

The diagnosis of SFN is based on the presence of suggestive clinical symptoms (pinprick and thermal sensory loss or hyperalgesia or allodynia) together with abnormal warm or cold threshold, or both, at the foot assessed by Quantitative sensory testing (QST); skin biopsy may be used in selected cases. The diagnosis of SFN necessitates abnormalities of at least two methods as measured by skin biopsy, QST, and the quantitative sudomotor axon reflex test (QSART) [15,16].

To date, there are no specific treatments for small fibres neuropathies. Therapy is essentially focused on the underlying disease that caused the neuropathy associated with symptomatic treatment [17].

### 2.2. Methods of Evaluation of Small Fiber Neuropathies

Functional impairment of small fibers (Aβ-fibers, and C-fibers) can be assessed using different neurophysiological methods, those currently most used are summarized below:QST is a well-established method that evaluates both pain and loss of sensory function. However, although this method is widely used, on the other hand, there are several limitations. First, the results are quite variable since is a psychophysical test, second is not localizing the lesion (central vs peripheral), third is time-consuming [18,19].Nociceptive evoked potentials are methods based on the selective activation of A delta and C fibers. There are several methods available: the radiant heat used for laser-evoked potentials (LEPs) [20,21,22,23,24,25,26], the contact heat for contact-heat-evoked potentials (CHEPs) [27,28,29], and the electrical skin stimulation for pain-related evoked potentials (PREPs) [30,31,32].Autonomic nervous system testing may be useful in the syndromic workout as autonomic dysfunction is often present in SFN. It may be detected by several techniques testing of sudomotor function [33]. Among the most used methods to study sudomotor function, are the Thermoregulatory Sweat Test (TST), the quantitative sudomotor axon reflex test (QSART), the quantitative direct and indirect axon reflex testing (QDIRT), the sympathetic skin response test (SSR), electrochemical sweat conductance methods with the SUDOSCAN device (Impeto Medical: Paris, France) [34,35,36,37].

All such methods require complex, long, and frequently invasive examinations so the simple, fast, and patient’s friendly method of providing information on small nerve fibers’ status would be of great importance.

### 2.3. Corneal Nerve Fibers

The cornea is the most important component of the eye’s optical system with its unique anatomical and physiological properties of transparency and avascularity [38]. The corneal intense sensory innervation derives from the ophthalmic branch of the trigeminal nerve and ensures the reflex function and regulates tear production and blinking. It is responsible for maintaining ocular surface homeostasis, mandatory for corneal health and function through the release of neurotrophic factors that promote epithelial healing and maintain the structural and functional integrity of the ocular surface [39,40,41].

Corneal sensory nerves reach the corneoscleral limbus with the anterior ciliary nerves and form the circumferential limbal plexus from which nerve fibers penetrate radially into the corneal stroma [41,42,43,44]. Then they run forward in the anterior third of the stroma towards the center, parallel to collagen lamellae, losing their myelin sheath, but remaining encased in Schwann cells not altering the corneal transparency and forming the mid-stromal nerve plexus in the anterior third of the stroma [40,41,42,43,44].

The fibers turn vertically towards the surface and penetrate the Bowman membrane to form the so-called subbasal nerve plexus (SBNP) that runs parallel to the corneal surface below the epithelial layer. The SBNP fibers direct centripetally converging inferiorly to the corneal apex with a whorl-like pattern located about 2.51 ± 0.23 mm inferonasal [41,42].

Sensory nerves that are peripheral branches of trigeminal neurons with small myelinated (A-delta) or unmyelinated (C) axons form the human corneal innervation [40,41,42,43,44].

Polymodal nociceptors are the most frequent, with a prevalence of 70%, and are constituted by C-type nerves. Twenty percent are mechanoreceptors formed by myelinated A-delta fibers, and the remaining 10% are cold receptors composed of Aδ and C fibers [38,39,41,43,44].

Corneal sensory nerves deliver the perception of thermal, mechanical, and chemical stimuli to the brain and induce the blink reflex, mediated by the afferent V nerve branches and the efferent VII nerve’s secretory and motor branches to ensure ocular surface protection. The sensory nerves provide the afferent impulse for reflex tearing, which involves parasympathetic nerves promoting tear production and secretion [38,39,40,41,42,43,44].

Autonomic innervation is represented predominantly by sympathetic nerve fibers. They originate from the superior cervical ganglion and are mixed with the sensory nerves in the perilimbal and limbal plexus. Autonomic fibers regulate the secretion of the tear film components by goblet cells, lacrimal gland, and meibomian glands. In the limbal stroma, the sympathetic fibers are located close to the vessel and have a vasomotor role [38,40].

### 2.4. Corneal Confocal Microscopy

IVCM is a tool which is used in the clinical investigation of the cornea in terms of health and disease. It is a non-invasive method, and it allows for the visualization of all corneal layers, including cells and nerves, with a comparable resolution to histology.

The principle that enables optical sectioning of the cornea is based on the so-called “common focal point”. Illumination and detection paths are located on the same focal plane due to the optical system that links the light rays focused on the tissue with those reflected and captured by the objective lens, so the term confocal was adopted. At the same time, since the light is focused on a restricted examined area, the light reflected from structures outside the focal point is discarded.

Three different types of confocal microscopes are in use to investigate in vivo corneal structures and several discrepancies in image quantification arise due to the differences in instruments design and functioning [3,4,45,46,47,48,49,50,51,52,53,54,55,56,57].

Tandem Scanning Confocal Microscope (TSCM) captures images using a rotation disc. It has a narrow depth of field of 11 microns and a small aperture that makes difficult the visualization of the small structures with reduced brightness and low contrast resulting in poor resolution of the nerve imaging [44].TSCM is no longer in production but in the literature, we can find several papers reporting confocal studies performed with this device. Slit Scanning Confocal Microscope (SSCM) is based on a light scanning slit, it works with a wider aperture and increased depth of field compared to the previous device, it is also faster and has better contrast and resolution, which determines high-quality images [44,45]. The images obtained with SSCM exhibit a higher contrast, brightness, sharpness, and more details that allow visualization of the SBNP with a high resolution [41,45]. SSCM is mainly represented by Confoscan 4 (Nidek Technologies, Padova, Italy) and ConfoScan P4 (Tomey, Erlangen, Germany) that together to the Laser Scanning Confocal Microscope (LSCM) represented by HRT with Rostock Cornea Module (Heidelberg Eng GmbH) constitute the most used systems worldwide. LSCM uses a coherent, high-intensity laser source and provides very detailed images of the anterior cornea being less accurate in the examination of the posterior layers [45]. High contrast and resolution that characterize this device offer high-quality images [45,53,55].

In IVCM the normal SBNP fibers appear as straight and beaded, well-defined, and clearly identified reflective structures. Figure 1.

The radial arrangement of the nerve fibers converges toward the central area to form a whorl-like pattern l about 1–2 mm below the center. The whorl was represented in the reconstructed wide field map obtained by the mapping of total confocal SBNP images, and the nerve fiber density is higher in the whorl when compared to the center [46,47].

Although the IVCM is an excellent tool to visualize the SBNP fibers with a high resolution, it is not able to provide images of the corneal epithelial terminals due to their small size [42,43].

The main parameters considered for morphological and densitometric analysis of the subbasal nerves are corneal nerve fibers length (CNFL) related to the total length of nerve fibers within an image and expressed in mm/mm^2^, corneal nerve fibers density (CNFD) (fibers/mm^2^) responding to the total number of fibers per frame divided by the area of the frame in mm^2^, corneal nerves branching density (CNBD) (n/mm^2^) reporting the total number of branches divided by the area of the frame in mm^2^, Tortuosity coefficient (TC) reflects the variability of nerves direction, frequency, and extent of changes in direction of nerve fibers, expression of neural degeneration and regeneration. Tortuosity increment, a sign of nerve damage, indicates the regeneration occurring in damaged nerves with active fibers growth [46,47,48,49,50,51,52,53,54].

Additionally, the beading frequency (BF) (n/mm^2^) represents the number of beads and hype-reflective dilatations along the nerve fibers in relation to axonal and sensory terminals containing glycogen and mitochondria that are considered indicative of metabolic activity [47,48,49,50,51,52,53,54,55,56].

The CNFL is considered a more stable parameter and it is diminished in different corneal conditions. It was determined as the optimal and the most reliable parameter for the detection of diabetic sensorimotor polyneuropathy [1,57,58].

Nowadays, different techniques are used to quantify the SBNP parameters, and they comprise manual, semi-automated, and fully automated methods of nerve tracing and analysis [45,46,47,48,49,50,51,52,53,54,55,56,57,58].

The normal values of SBNP density and morphology reported in the literature vary significantly in relation to the different methods and software applied for the analysis. Accordingly, to the different reports available, the CNFD obtained with Tandem Scanning Confocal Microscope (TSCM) varies from 5867 ± 3.31 to 8404 ± 2.012 mm/m^2^ with the Slit Scanning Confocal Microscope (SSCM) reports it varies from 26.5 ± 7.5 to 68.08 ± 20.3 fibers/mm^2^, and for the Laser Scanning Confocal Microscope (LSCM) the normal values for the CNFD reported are 23.3 to 37.2 ± 6.7 fibers/mm [2,4].

As to the CNFL the SSCM report values from 6.1 ± 1.2 to 13.5 ± 0.3 mm/mm^2^ and LSCM the range from 16.1 to 26.4 mm/mm^2^. Relatively to the CNBD, the SSCM values range from 25.4 ± 3 to 78.9 ± 30.4 branches/mm^2^ and those obtained with LSCM vary from 30.6 to 92.7 ± 38.6 branches/mm^2^. The data relating to the normal beadings frequency range from 36.96 ± 9.9 to 222 ± 43 beads/mm^2^ and the tortuosity coefficient from 1.09 ± 0.54 to 2.2 ± 0.9, whereas the nerve thickness range from 0.52 ± 0.27 to 2.9 ± 0.2 mm [6,38,41,46,47,48,49,50,51,52,53,54]. The relevant discrepancies of the normal values reported in the literature confirm the need for the use in comparative and observational studies, of the same type of confocal microscope and to enroll a healthy normal age-matched control group for statistical purposes.

It can be summarized that the nerves density decreases with age and, according to existing reports, this change varies from 0.25% to 0.9% per year, and there are no differences between eyes and between males and females whereas the tortuosity increases with age of 0.044 per year for men and 0.046 per year for women [54,56].

IVCM allows fast, non-invasive, and repeatable examination of the corneal SBNP with the possibility of immediate qualitative and quantitative analysis.

Due to these characteristics and information provided the IVCM is recognized as a fundamental tool to investigate the status of the nerve fibers in local and systemic diseases and its diagnostic role in small fibers polyneuropathies is remarkable and CNFL is considered a biomarker of peripheral nerve fiber degeneration [50].

In this review, we aimed to summarize the results of IVCM in small fibers polyneuropathies that originate from different underlying diseases to assess the validity of this investigative tool that may replace the invasive, long, and complex neurological test for small fibers status assessment.

Additionally, for each type of SFN, we evaluated the correlation between the nerve density and morphology with some parameters such as disease duration, disease severity, use of medical therapies, and skin biopsy.

### 2.5. Database and Literature Search

We used PubMed and Medline databases for the literature search. Search for “corneal confocal microscopy and small fibers polyneuropathies”, “corneal subbasal nerve plexus and small fibers polyneuropathies”, “confocal microscopy and corneal neuropathy“, “corneal nerves and small fiber polyneuropathies”, “confocal microscopy and peripheral small fibers polyneuropathy”, “corneal nerves assessment in neurological disease”, “refractive surgery and small fibers polyneuropathies”, “cataract surgery and small fibers polyneuropathies”

### 2.6. Inclusion and Exclusion Criteria

We included original articles, reviews, and brief reports that contributed information relevant to this review. We included only studies in vivo with confocal microscopy on humans. We considered publications in English with no limits for the period of publication.

We excluded abstracts, letters, conference proceedings, case reports, experimental studies, and post-mortem studies.

## 3. Results

The overall literature data are represented in Table 1, Table 2, Table 3, Table 4 and Table 5.

Parkinson’s disease results as the most represented with 10 studies, followed by Sjogren disease (6), fibromyalgia (4) and multiple sclerosis (3). Other small fibers disorders are reported in 17 papers with single studies or up to 2 papers.

### 3.1. Parkinson’s Disease

Table 1 shows IVCM findings in Parkinson’s disease obtained with Nidek-ConfoScan 4 and 1 HRT2 and 3. Nerve parameters were evaluated with different available software such as ACCmetrics, NeuronJ, and Navis.

**Table 1 jcm-12-00664-t001:** Corneal SBNP parameters in patients with Parkinson’s disease.

Authors	Number of Patients	CNFL	CNFD	CNBD	TC	BF	CCM	Localization	Method Used	Main Findings
[Ref.]	(Parkinson/Healthy)						Used			
Andreasson et al. [57].	42/13	x		x			HRT 3	Central	Automated	No group differences between CNFL (*p* = 0.81) and CNBD (*p* = 0.91).
	(21: Parkinson + Restless legs Sindrom; 21: Parkinson)									
Kass-Iliyya et al. [58].	26/26	x	x	x			HRT 3	Central	ACCmetrics Automated	Reduction of CNFD in PD (*p* = 0.003).
										Increase of CNBD (*p* < 0.001) and CNFL (*p* = 0.031) in PD.
Lim et al. [59].	98/26	X	X	X			HRT 3	Central	ACCmetrics Automated	Reduction of CNFD (*p* = 0.001), CNBD (*p* = 0.003), and CNFL (*p* = 0.001) in participants with PD compared to controls
Arrigo et al. [60].	3/0				x	x	Nidek-ConfoScan 4	Central	NAVIS software	Increase of beading (BF). No significant changes in density, except for a reduction shown by the left eye of PD2.
Podgorny et al. [61].	26/22	x	x	x			HRT 3	Central	ACCmetrics Automated	Reduction of CNFL (*p* = 0.013), (*p* = 0.013), CNFD, in patients with PD. CNFD
										didn’t have a significantly difference. (*p* = 0.058)
Misra et al. [59].	15/15		x				HRT 2	Central	Automated	Reduction of CNFD in patients with PD compared with controls. (*p* < 0.0001)
Reddy et al. [62].	(12: 7 patients with Progressive Sopranuclear Palsy (PSP) and 4 patients with PD)		x				Nidek-ConfoScan 4	Central	NeuronJ	There were no differences in corneal sub-basal nerve density between the 3 groups.
Daggumilli et al. [63].	120/30		x				Nidek-ConfoScan 4	Central	NeuronJ	Reduction of CNFD in PD amantadine and PD amantadine naive group compared with healthy controls after 1-year follow-up. (*p* = 0.032 and 0.048)
	150 subjects: 90 PD with amantadine, 30 PD naïve amantadine, 30 controls)									
Anjos R. et al. [64].	25/25	x	x	x	x		HRT 2	Central	Automated	Corneal nerve fiber morphology differed in both groups, with a lower global fiber density, branch density, and higher tortuosity in Parkinson patients (*p* < 0.05). These parameters were found to be related to dopaminergic medication exposure.
Avetisov S.E. et al. [65].	16/0			x	x	x	HRT 3	Central		Increase in a number of branches from the main nerve trunks, an increase in the tortuosity of CNF, and a “beaded” shape.

CNFL was measured in 6 studies and it was significantly reduced in 3 papers (Figure 2), unvaried in 2, and increased in one. CNFD was analyzed in 8 studies and it was reduced in 6 papers being unvaried in two. CNBD changes were analyzed in 6 papers and the findings are rather conflicting with 3 reduced, 2 increased and 2 unvaried. Three studies evaluated TC and confirmed a significant increase in tortuosity. Only 2 reports considered BF and confirmed a significant increase in this parameter [59,60,61,62,63,64,65,66,67,68].

### 3.2. Sjogren Syndrome

Two authors used an HRT3 confocal microscope, two Nidek-ConfoScan 4, one Heidelberg Retina Tomograph II, and one Model 165A (Tandem Scanning Corp., Reston, VA, USA). Two studies highlight a significant reduction of CNFL in patients with SS compared with healthy groups. Regarding the CNFD, four authors report a statistically significant reduction in patients with SS, one found a significant increase, and one no differences between the two groups. One study shows no differences in the CNBD parameter between SS and healthy patients. A significant increase in TC in SS patients is reported in 3 studies. No studies evaluate BF in SS patients [69,70,71,72,73,74].

**Table 2 jcm-12-00664-t002:** Corneal SBNP parameters in patients with Sjogren syndrome.

Authors	Number of Patients	CNFL	CNFD	CNBD	TC	BF	CCM	Localization	Method Used	Main Findings
[Ref.]	(Sjogren/Healthy)						Used			
Bercelos F. et al. [69].	−55: SjS;	x	x		x		Heidelberg!Retina Tomograph II,	Central	ImageJ	Reduction of CNFL and CNFD (*p* < 0.001).
	−63 Sicca;									-Increase TC.
	−18: rheumatoid arthritis (RA).									
	20: healthy controls									
Tuominen et al. [70].	10/10		x				model 165A, Tandem Scanning Corp., Reston, VA, USA	Central		No difference was noted in nerve density.
McNamara et al. [71].	10/10	x	x	x			Nidek Confoscan 4	Central	Nerve tracking V1.0	Reduction of CNFD (*p =* 0.03) and CNFL (*p* < 0.05) in SS patients compared to controls.
Levy et al. [72].	30/10		x				HRT 3	Central	ImageJ	CNFD was significantly increased (*p* < 0.0001) associated with a decrease in DC density (*p* < 0.0001).
	30:patients with SS treatment treated with CsA 0.05% twice daily for six months									
Tepelus et al. [73].	44/10		x		x		HRT 3	Central	NeuronJ	Reduction of CNFD (*p* < 0.001) and increase of TC (*p* < 0.05) in patients with SS syndrome
Villani et al. [74].	35/20		x		x		Nidek Confoscan 2	Central	ImageJ	Reduction of CNFD (*p* < 0.001) and increase of TC (*p* < 0.0001)

### 3.3. Fibromyalgia

Three studies used a ConfoScan 4, with NAVIS software being in two papers and ImageJ in one study. One paper reported HRT 3 and the ACC metrics automated analysis of data.

All these studies evaluated CNFD and agreed on a significant reduction of this parameter in affected patients, whereas one report provides additional data on CNFL and CNBD significant reduction in patients with fibromyalgia [73,74,75,76,77,78] Table 3.

**Table 3 jcm-12-00664-t003:** Corneal SBNP parameters in patients with fibromyalgia.

Authors	Number of Patients	CNFL	CNFD	CNBD	TC	BF	CCM	Localization	Method Used	Main Findings
[Ref.]	(Fibromyalgia/Healthy)						Used			
Erkan Turan K. et al. [75].	34/42		x				Nidek-ConfoScan 3	Central	ImageJ	Total nerve density, long nerve fibers, and the number of nerves were all lower in patients with FM compared with controls (*p* < 0.001).
Ramirez et al. [76].	7/7		x				Nidek-ConfoScan 4	Central	NAVIS software	Reduction of CNFD (*p* = 0.02)
Ramirez et al. [77].	28/0		x				Nidek-ConfoScan 4	Central	NAVIS software	Reduction of CNFD (nerve density < normality cutoff point found in their previous study).
	(15: Fibromyalgia + anxiety or depression; 13: Fibromyalgia without depression)									No difference between two groups.
Oudejans L. et al. [78].	39/0	x	x	x			HRT 3	Central	ACCmetrics Automated	CNFL was significantly decreased in 44% of patients compared to age-
										and sex-matched reference values; CNFD and CNBD were significantly decreased
										in 10% and 28% of patients.
										CNFL values correlated with CNBD (Pearson’s r = 0.81) and CNFD (Pearson’s r = 0.90) (*p* < 0.01)

### 3.4. Multiple Sclerosis

Three studies compared central cornea confocal findings in patients with Multiple Sclerosis (MS) and healthy patients. (Table 4)

**Table 4 jcm-12-00664-t004:** Corneal SBNP parameters in patients with multiple sclerosis.

Authors	Number of Patients	CNFL	CNFD	CNBD	TC	BF	CCM	Localization	Method Used	Main Findings
[Ref.]	(SM/Healthy)						Used			
Bitirgen et al. [79].	57/30	X	X	X			HRT3	Central	ImageJ	Reduction of CNFL (p: 0.001), CNFD (p: 0.002) and CNBD (p: 0.001)
Mikolajczak et al. [80].	26/26		X				HRT 3	Central	NeuronJ	Reduction of CNFD in MS patients compared to
										controls. (*p* = 0.007)
Petropouls et al. [81].	25/25	X	X	X			HRT 3	Central	ACCmetrics Automated	Reduction of CNFD (*p* < 0.0001), CNFL (*p* < 0.0001), and CNBD (*p* = 0.0003) compared with controls.

All authors used an HRT3 confocal microscope and analysis was performed with three different methods. Two studies analyzed CNFL, CNFD, and CNBD and reported a significant reduction of all parameters in the MS group. One study analyzed only CNFD which was significantly reduced. No one of the studies analyzes TC and BF [79,80,81].

### 3.5. Other Pathologies

Table 5 reported the results for other pathologies for a total of 16 papers. In 14 studies HRT and two Confoscan 4 were used and the AAC metrics automated software was used in most of the analysis.

**Table 5 jcm-12-00664-t005:** Corneal SBNP parameters in patients with different diseases characterized by small fibers neuropathy.

Authors	SFN	Number of Patients	CNFL	CNFD	CNBD	TC	BF	CCM	Localization	Method Used	Main Findings
[Ref.]	Type	(SFN/Healthy)						Used			
Rousseau et al. [82].	TTR-FAP	15/15	X					HRT 3	Central	ImageJ	Reduction of CNFL (*p* = 0.02).
Zangh et al. [83].	TTR-FAP	15/15	X	X	X			HRT 3	Central/inferior whorl	ImageJ	Reduction of IWL (Inferior whorl length) (*p* = 0.006), CNFL (*p* = 0.005), CNBD (*p* = 0.008), and CNFD (*p* = 0.014).
Barnett et al. [84].	Neurofibromatosis I	52/0	x					HRT 2	ND	Automated	CNFL was below the normative level (10.1 ± 2.7 mm/mm^2^).
Bitirgen et al. [85].	BEHCET	49/30	x	x	x			HRT 3	Central	ACCmetrics Automated	Reduction of CNFD (*p* = 0.001) and CNFL (*p* = 0.031) in patients with Behcet.
											CNBD did not differ significantly (*p* = 0.067).
Bucher et al. [86].	SNF	14/14		x		x		HRT 2	Central	ImageJ	Reduction CNFD (*p* = 0.001).
											Increase of TC (*p* = 0.5).
Tavakoli et al. [85].	Fabry’s Disease	ND	x	x	x	x		HRT 3	ND	CCM Image Analysis Tools v 0.6	Nerve damage in idiopathic small fibre neuropathy and Fabry disease.
Ferrari et al. [3].	SLA	8.89	x	x		x		HRT 2	ND	Automated	Reduction of CNFD (*p* < 0.011) and CNFL (*p* = 0.004) in SLA patients.
											Increase TC in SLA patients (*p* < 0.0005).
Fu et al. [87].	SLA	66/64	x	x	x			Nidek-ConfoScan 3	Inferior whorl	ImageJ	Reduction of CNFL (*p* < 0.05) and of CNFD (*p* = 0.011). Increase of CNBD (*p* = 0.040).
Gad et al. [88].	Celiac Disease	20/20	x	x	x	x		HRT	Central and inferior whorl	ACCmetrics Automated	CNFL (*p* = 0.8), CNFD (*p* = 0.5), CNBD (*p* = 0.1) did not differ between children with Celiac Disease and controls.
											Reduction of TC (*p* = 0.01).
Kemp et al. [89].	HIV^+^	20/20	x	x	x	x		HRT	Central	ACCmetrics Automated	Reduction of CNFD (*p* < 0.001), CNBD (*p* = 0.01), and CNFL (*p* = 0.001).
											Increase of TC (*p* = 0.03).
Khan et al. [90].	Diabetes + Charcot	20/20	x	x	x			HRT3	Central	ACCmetrics Automated	Reduction of CNFD (*p* < 0.001), CNBD (*p* < 0.01), and CNFL (*p* < 0.001).
O’Neill et al. [91].	Burning mouth Syndrome (BMS)	17/14	x	x	x			HRT3	Central	ACCmetrics Automated	Reduction CNFD (*p* = 0.007), CNFL (*p* = 0.007).
											There was no difference in CNBD BMS vs Controls (*p* = 0.06).
Schneider et al. [92].	Chronic inflammatory demyelinating polyneuropathy (CIDP)	16/15	x	x	x	x		HRT3	Central	ImageJ	Reduction CNFD (*p* < 0.0001), CNFL (*p* < 0.001), CNBD (*p* < 0.0001).
											Increase of TC (*p* < 0.01)
Sturniolo et al. [93].	Wilson Disease	24/24		x	x	x		Nidek-ConfoScan 4	Central	Nerve tracking V1.0	Reduction of CNFD (*p* < 0.0001), CNBD (*p* < 0.0001).
											Increase of TC (*p* < 0.001).
Culver et al. [94].	Sarcoidosis	48/16	X	X	X	X	X	HRT 3	Central	ACCmetrics Automated	-Double-blind, randomized, placebo-controlled.
		(16: CIBINETIDE 1MG									-Reduction CNFA: 1 mg and placebo groups (*p* = 0.748 and *p* = 0.32).
		16: CIBINETIDE 4MG									-Increase CNFA: 4 mg and 8 mg groups (*p* = 0.084 and *p* = 0.274).
		16: CIBINETIDE 8MG									
		16: PLACEBO)									
Pagovich et al. [95].	Friedreich Ataxia	23/14	x	x	x			HRT 3	Central	ACCmetrics Automated	Reduction of CNFD
											(*p* < 0.0001) e CNFL (*p* < 0.002) in FRDA
Tavakoli et al. [96].	Charcot-Marie-Tooth Disease type 1A patients	12./12	x	x	x			HRT 3	Central	ACCmetrics Automated	Reduction of CFND (*p* = 0.01), CNBD (*p* = 0.02), and CNFL (*p* = 0.0001) in CMT1A patients compared with control subjects

CNFL was measured in 14 studies, and it resulted significantly reduced in 13 papers and unvaried in one study performed on children with celiac disease.

CNFD was studied in 14 papers, and it was significantly reduced in 13 being as well unvaried in the same pediatric group.

CNBD is reported in 12 papers with 9 reduced, and 3 unvaried: in burning mouth syndrome, celiac pediatric population, and Behcet disease. It resulted in an increase in one paper on SLA.

Tortuosity was analyzed in 7 studies with a significant reduction in all reports.

Beading frequency was not considered [80,81,82,83,84,85,86,87,88,89,90,91,92,93,94,97].

### 3.6. Correlation of Disease Duration and Nerve Alterations

Zangh et al. reported that in transthyretin familial amyloid polyneuropathy (TTR-FAP) patients all CCM parameters were significantly reduced with disease progression. IWL (Inferior whorl length) (*p* = 0.006), CNFL (*p* = 0.005), CNBD (*p* = 0.008), and CNFD (*p* = 0.014) were significantly lower in early-phase patients [84].

When investigated, no correlation was found between corneal nerve measurements with disease duration in Parkinson’s disease (PD) and Fibromyalgia. As well as between corneal nerve fiber measures and the duration of HIV [1,76,89].

### 3.7. Correlation of Disease Severity and Nerve Alterations

In most of the studies, there was a significant correlation between the severity of the disease and the alterations of the nerves at the CCM. Especially Rousseau et al. reported that CNFL correlated with the severity of both the sensorimotor and autonomic neuropathy in TTR-FAP [82].

Both IWL (Inferior whorl length) and CNFL correlated with the severity of neuropathy in TTR-FAP patients according to Zangh et al. [83].

Reduction of CNFD according to increasing severity of neuropathy in fibromyalgia (*p* = 0.02) and the degree of corneal denervation correlated with small fiber neuropathy symptom burden, with dysautonomia symptoms, and with one Fibromyalgia Impact Questionnaire domain [77].

### 3.8. Correlation of Medical Therapy and Nerve Alterations

Culver et al., rated the effect of Cibinetide on the corneal nerve fiber area (CNFA) and regenerating intraepidermal fibers (GAP-43þ) in subjects with sarcoid-associated SNFL. Cibinetide significantly increased small nerve fiber abundance in the cornea (CNFA) compared to placebo for the 4 mg group (*p* = 0.084). Further, changes in CNFA correlated with changes in GAP-43þ (q 1⁄4 0.575; P 1⁄4 0.025) and 6MWT (q 1⁄4 0.645; P 1⁄4 0.009) [94].

No correlation was found between corneal nerve fiber parameters and the duration of treatment with antiretroviral medication in HIV patients [98]. While Andréasson et al. reported an association between CNFL and CNBD and the duration of L-dopa therapy in subjects with Parkinson’s disease (ρ = −0.36, *p* = 0.022) [66].

### 3.9. Correlation between Skin Biopsy and Nerve Alterations

In all studies in which the authors compared the results of nerve alterations in IVCM with the results of skin biopsy, there was a statistically significant positive correlation.

Rousseau et al. demonstrated that patients with TTR-FAP with altered sensory nerve action potentials and intraepidermal nerve fiber density had a shorter CNFL [82].

The publication of Kass-Iliyya et al., shows both CNFD and IENFD (intraepidermal nerve fiber density) were significantly lower in PD patients compared to controls (r = 0.464) (*p* = 0.026) [67].

## 4. Discussion

Corneal sensory innervation is well-developed in humans, and it is provided by the trigeminal nerve. Proper innervation is important for corneal tropism, healing capacity, and maintenance of ocular surface homeostasis.

Since the sub-basal nerve plexus fibers are involved in different systemic and neurological diseases, the possibility to investigate nerve morphology in vivo offers a unique opportunity to explore the small fibers within the cornea. Specifically, in the course of SF diseases, the corneal nerves undergo alterations similar to those occurring in the other tissues, and IVCM enables us to detect such changes in a non-invasive way.

Indeed, corneal confocal microscopy is an in-vivo, non-invasive, and reproducible diagnostic technique that allows the examination of the living human cornea in healthy and pathological situations.

Specifically, IVCM provides us with direct information on the health of the sub-basal nerve plexus and corneal and ocular surface conditions. IVCM is also used to study the peripheral nerve in different systemic diseases. As the IVCM permits the immediate analysis of the corneal nerves it became, over time, an important tool for diabetic peripheral neuropathy and autonomic neuropathy assessment and follow [1,2,3,4,5].

In this review, we analyzed reports concerning corneal nerves investigation in SF polyneuropathies, summarizing the data related to the nerve’s morphology and density changes.

We have summarized the current knowledge about corneal nerve involvement in different SFN with different underlying systemic diseases.

In almost all studies, the main finding was a significant reduction in the CNFL, CNFD, and CNBD as well as an increased tortuosity and altered beadings.

In Parkinson’s disease, not all studies agree on the nerve alterations even if the majority of reports confirm a significant reduction of all considered parameters suggesting a small fibers involvement. However, there is no consensus on the underlying pathophysiology of peripheral neuropathy in PD.

Cumulative Levodopa exposure has been indicated as responsible for large fiber neuropathy via homocysteine accumulation and reduced levels of folate as well as vitamin B12. [99] In addition, since there is no relationship between peripheral denervation and Levodopa exposure it is like that peripheral nerve involvement may be an intrinsic feature of the disease, especially about small fiber neuropathy [100] Future studies are needed to elucidate this point.

Also the pathogenesis of SFN involvement remains to be discovered in multiple sclerosis although an autoimmune etiology is thought to be the most likely etiology in the majority of idiopathic SFN cases [98].

Interestingly in Fibromyalgia, all studies report a significant reduction of all nerve parameters (CNFD, CNFL, and CNBD) although this is still awaiting confirmation [101].

However, the discrepancies in results obtained in PD and SS might be probably attributed to the different clinical profiles of enrolled patients. Such a hypothesis was raised by Che and Yang for PD patients in their article on usefulness of IVCM in PD-associated neuropathy [5].

The high concordance of data puts this finding as highly significant in the diagnosis and monitoring of nerve status in fibromyalgia and MS disease.

Considering the group of other SF pathologies summarized in Table 5, the common finding of reduced nerve parameters is evidenced with only one exception in the pediatric population with celiac disease. Such data might be attributed to the young age of studied patients and future studies are needed as at the time; this is the only report considering pediatric patients.

The common finding in all analyzed studies on adults is a significant increase in nerve tortuosity that indicates nerve regeneration. Such data is suggestive of nerve suffering, distress, and regeneration and is common in all SF neuropathies independently from the main disease.

Only a few studies explored in detail the correlation between nerve parameters and disease duration.

Zangh et al. reported a strict correlation between nerve changes and the duration of the disease in transthyretin familial amyloid polyneuropathy (TTR-FAP) patients. Such a correlation was not found in fibromyalgia, HIV, and Parkinson’s disease [1,77,83,89]. We do believe that this important issue should be better investigated on a significantly greater number of subjects and in different SF polyneuropathies. Concerning the relationship between clinical signs and nerve alterations in the majority of the studies, there was a significant correlation between the severity of the disease and the alterations in the nerves detected by IVCM.

The other important evidence is the possibility of monitoring the effects of therapies on corneal nerve fibers using the IVCM. Culver et al. reported a significant positive correlation between medical therapy with Cibinetide and nerve density in sarcoid-associated SNFL [94].

Similarly, the association between CNFL and CNBD changes and the duration of L-dopa therapy in subjects with Parkinson’s disease was reported by Andréasson et al. [1].

An important strength point of IVCM is the sensitivity of examination. Bitirgen et al., postulated that IVCM could be considered an imaging biomarker for the detection of fibers loss in patients with MS [78].

Podgorny et al. recognized the IVCM as an effective tool for early diagnosis of fibers alterations in PD as the corneal nerves fibers changes precede those occurring in the cutaneous fibers. When no changes were observed in IENFD examination the CNBD and CNFL were significantly decreased [60].

The preliminary reports suggest that IVCM might be more sensitive than skin punch biopsy in detecting early nerve fiber damage or getting an early sign of nerve fiber regeneration.

Additionally, IVCM performed in association with PREP examination increased the identification of patients with small fibers impairment to 85% while QST, QSART, and proximal IENFD showed significantly lower impact [102].

Such findings open new perspectives on non-invasive diagnosis of SF neuropathies.

IVCM is currently applied in diabetology and neurology where the SF peripheral neuropathy may be early diagnosed and monitored in vivo with direct observation of the nerve status. Furthermore, the IVCM allows assessing the benefits of different specific therapies on nerve fiber regeneration, although, there are only a limited number of studies focusing on the effects of different therapies on SBNP in SF neuropathies.

Finally, the rapidly developing use of artificial intelligence in ophthalmology now permits the analysis of data with automated software in the context of screening ocular diseases and detecting early alterations with high sensitivity and specificity. The machine learning used in IVCM data analysis could provide precise information by using algorithms for fully automated nerve assessments to detect small fiber alterations [103,104].

## 5. Conclusions

IVCM is a fast-non-invasive ophthalmic imaging technique that can explore small nerve fibers in the cornea in a non-invasive way. IVCM has comparable if not superior diagnostic utility to intraepidermal nerve fiber density and neurophysiology for diagnosing SFN and the corneal SBNP fibers loss is associated with disease progression in different SFN.

Therefore, IVCM should be considered a crucial marker of neurodegeneration and regeneration and a surrogate end-point in clinical trials of new therapies in peripheral and central neurodegenerative diseases in addition to neurophysiological techniques and clinical parameters.

## Figures and Tables

**Figure 1 jcm-12-00664-f001:**
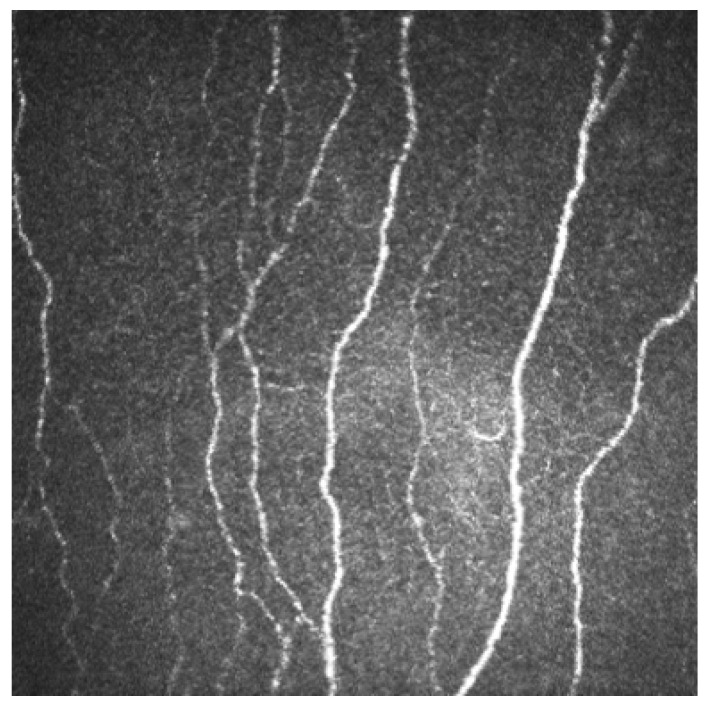
IVCM shows a normal healthy subbasal nerve fibers pattern. (400 × 400 microns).

**Figure 2 jcm-12-00664-f002:**
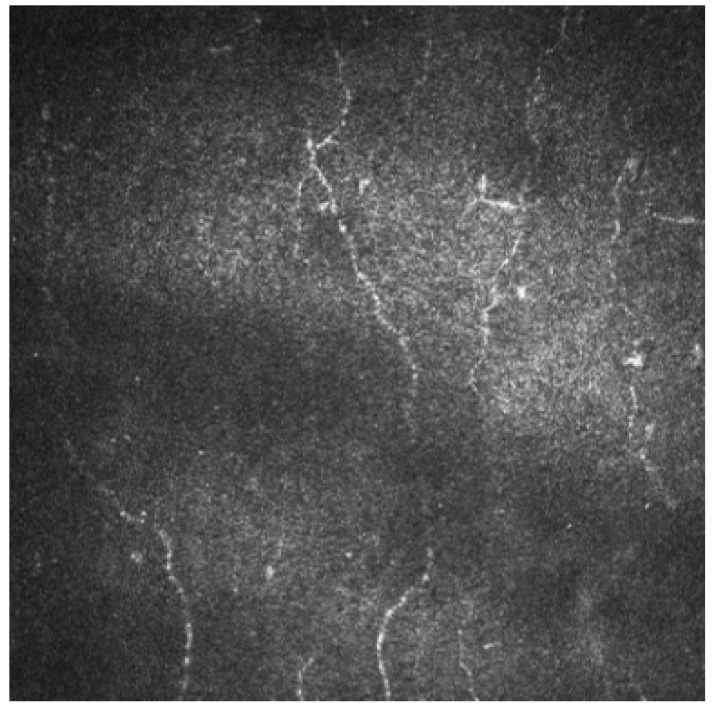
IVCM shows highly altered subbasal nerve fibers pattern in a patient with advanced Parkinson disease. (400 × 400 microns).

## Data Availability

Not applicable.

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
