# Peer review of "Corneal Sub-Basal Nerve Plexus in Non-Diabetic Small Fiber Polyneuropathies and the Diagnostic Role of In Vivo Corneal Confocal Microscopy"

_jcm, 2023, doi:10.3390/jcm12020664_

Round 1

Reviewer 1 Report

The evaluation of corneal fibers is becoming an attractive method for evaluation of peripheral small fiber neuropathies. Articles clarifying the advantages and limitations of the method are necessary. Several important issues emerge from the reading of this paper:

First It would be important modify the title: Confocal sub basal nerve plexus  “in non diabetic small fiber neuropathy” is a possibility. Perhaps a better option would be: “in different neurological affections” considering the results.

It should be clearly stated that Small fiber neuropathy is not a disease but rather a syndrome that has many etiologies and very likely more than one pathogenesis.

If the objective is to present a review of corneal confocal microscopy it is not clear why the authors dedicated almost two pages to the description of other methods of evaluation of small fiber neuropathies. This part could be shortened by referring the reader to recent articles on the subject.

The authors state that hyperlipidemia is a frequent cause of SFN. This is not an established issue. Metabolic syndrome has been postulated as an important contributing factor for the development of diabetic neuropathy, but it is not the same for hyperlipidemia itself. Recently D´Onofrio et al. using confocal microscopy, have described findings compatible with small fiber neuropathy in relation to hypertriglyceridemia; further studies are needed to confirm hypertriglyceridemia as cause of small fiber neuropathy, and even more studies are required to know its frequency (D’Onofrio et al., J Clin Lipidol 2022 Jul-Aug;16(4):463-471. doi: 10.1016/j.jacl.2022.04.006).

As the authors state, once entered into the cornea the myelinated fibers lose their myelin sheet and therefore the cornea itself only has C fibers. A delta myelinated fibers are only present in the peripheral cornea and in the nerves beyond the cornea. The paragraph starting with “the human corneal nerves…” may lead to confusion and must be clarified. Additionally A delta fibers cannot be qualified as “large.” This is a term reserved for A beta fibers. It is better to name them as “small myelinated fibers.”

The most surprising finding of the results and discussion is that three out of four pathologies listed as main source of articles, are not recognized as causing small fiber neuropathy. The authors need to discuss in much more depth what would be the meaning of the abnormalities described in the cornea in CNS diseases such as Parkinson´s Disease and Multiple Sclerosis. To state that “other small fiber disorders are reported in 17 papers…” accepting PD and MS as established cause of small fiber neuropathy is confusing and misleading. The papers the authors cite contain some discussion of what could be the meaning of these changes in the corneal fibers. These findings may even questioning that the evaluation of corneal fibers is a surrogate of peripheral small fiber disease. Regarding fibromyalgia, some publications have postulated that a group of patients with this diagnosis may have a small fiber neuropathy but this is still awaiting confirmation (Serra et al., Hyperexcitable C nociceptors in fibromyalgia. Ann Neurol. 2014;75:196-208. doi: 10.1002/ana.24065). The findings are interesting but go far beyond the simple diagnosis of small fiber neuropathy and it is fundamental to discuss their multiple potential significances. Otherwise the results and discussion are very confusing.

Author Response

Reviewer 1

Comments and Suggestions for Authors

The evaluation of corneal fibers is becoming an attractive method for evaluation of peripheral small fiber neuropathies. Articles clarifying the advantages and limitations of the method are necessary. Several important issues emerge from the reading of this paper:

 First It would be important modify the title: Confocal sub basal nerve plexus  “in non diabetic small fiber neuropathy” is a possibility. Perhaps a better option would be: “in different neurological affections” considering the results.

Authors response to reviewer:

We changed the title to : Corneal sub-basal nerve plexus in non diabetic small fibers polyneuropathies and the diagnostic role of the In Vivo Corneal Confocal Microscopy. We preferred “non diabetic” as not only neurological affections were included in this review.

It should be clearly stated that Small fiber neuropathy is not a disease but rather a syndrome that has many etiologies and very likely more than one pathogenesis.

Response to reviewer: it has now been clearly specified in the introduction

If the objective is to present a review of corneal confocal microscopy it is not clear why the authors dedicated almost two pages to the description of other methods of evaluation of small fiber neuropathies. This part could be shortened by referring the reader to recent articles on the subject.

Response to reviewer: as suggested this part has been significantly reduced to less than one page, referring the reader to other recent articles for further reading

The authors state that hyperlipidemia is a frequent cause of SFN. This is not an established issue. Metabolic syndrome has been postulated as an important contributing factor for the development of diabetic neuropathy, but it is not the same for hyperlipidemia itself. Recently D´Onofrio et al. using confocal microscopy, have described findings compatible with small fiber neuropathy in relation to hypertriglyceridemia; further studies are needed to confirm hypertriglyceridemia as cause of small fiber neuropathy, and even more studies are required to know its frequency (D’Onofrio et al., J Clin Lipidol 2022 Jul-Aug;16(4):463-471. doi: 10.1016/j.jacl.2022.04.006).

Response to reviewer: hyperlipermia has been replaced with metabolic syndrome

As the authors state, once entered into the cornea the myelinated fibers lose their myelin sheet and therefore the cornea itself only has C fibers. A delta myelinated fibers are only present in the peripheral cornea and in the nerves beyond the cornea. The paragraph starting with “the human corneal nerves…” may lead to confusion and must be clarified. Additionally A delta fibers cannot be qualified as “large.” This is a term reserved for A beta fibers. It is better to name them as “small myelinated fibers.”

Response to reviewer: the sentence has been changed as follows “Sensory nerves that are peripheral branches of trigeminal neurons with small myelinated (A-delta) or unmyelinated (C) axons form the human corneal innervation”

The most surprising finding of the results and discussion is that three out of four pathologies listed as main source of articles, are not recognized as causing small fiber neuropathy. The authors need to discuss in much more depth what would be the meaning of the abnormalities described in the cornea in CNS diseases such as Parkinson´s Disease and Multiple Sclerosis. To state that “other small fiber disorders are reported in 17 papers…” accepting PD and MS as established cause of small fiber neuropathy is confusing and misleading. The papers the authors cite contain some discussion of what could be the meaning of these changes in the corneal fibers. These findings may even questioning that the evaluation of corneal fibers is a surrogate of peripheral small fiber disease. Regarding fibromyalgia, some publications have postulated that a group of patients with this diagnosis may have a small fiber neuropathy but this is still awaiting confirmation (Serra et al., Hyperexcitable C nociceptors in fibromyalgia. Ann Neurol. 2014;75:196-208. doi: 10.1002/ana.24065). The findings are interesting but go far beyond the simple diagnosis of small fiber neuropathy and it is fundamental to discuss their multiple potential significances. Otherwise the results and discussion are very confusing.

We thank the reviewer for underlying this important point. Our assumptions in the current form could be misleading. Indeed, several studies have shown evidence of small and large fiber peripheral neuropathy in PD. However as correctly underlined there is no consensus on the underlying pathophysiology of peripheral neuropathy in PD. Several evidences suggest that peripheral nerve involvement may be an intrinsic feature of the disease, especially in relation to small fiber neuropathy. Similarly, we also tuned down in the discussion the sentence relative to fibromyalgia. These considerations have now been added in the text.

Reviewer 2 Report

In Vivo Confocal Microscopy IVCM could have  an increased interdisciplinary diagnostic and prognostic value (diabetes, neurology , rheumatology) if the interdisciplinary studies had more nuanced correlation regarding the severity of general conditions ( humoral parameters, evolution, progression of the disease, response to treatment, disabilities) on the one hand and on the other , if IVCM follows a quantified scale of injuries both statically and dynamically.

Author Response

Reviewer 2

In Vivo Confocal Microscopy IVCM could have an increased interdisciplinary diagnostic and prognostic value (diabetes, neurology , rheumatology) if the interdisciplinary studies had more nuanced correlation regarding the severity of general conditions ( humoral parameters, evolution, progression of the disease, response to treatment, disabilities) on the one hand and on the other , if IVCM follows a quantified scale of injuries both statically and dynamically.

We thank the Reviewer for this comment. Indeed, we believe that in the next future the use of IVCM and possibly a machine learning system could constitute a valid additional diagnostic tool.

Reviewer 3 Report

Summary of Manuscript: The manuscript reviews the literature on the use of In Vivo Confocal Microscopy of corneal innervation for small fiber neuropathies. They compare the outcomes of various clinical testing procedures in detecting SFN, and recommend IVCM as a simpler, less invasive yet effective diagnostic tool. 

Critique Points:

  1. There are minor issues with the language throughout the text , and while it is easily comprehensible, it is noticeable. IF would be worthwhile to copy and language check the entire manuscript. A few examples are given here: 
    1. The authors use the term “resume: as in the short version of a curriculum vitae but they more likely mean summarize.  This should be corrected throughout the text.
    2. Check the use of sensibility throughout the manuscript, and correct to sensitivity.
    3. Line 176 “stimulate postganglionic sudomotor unmyelinated nerves C-fibers. 
    4. Line 269. A whorl-like pattern 1 about 102mm inferiorly…
  2. It might be helpful for the authors to give more detail on the autonomic fibers (peripheral and limbus) versus trigeminal fibers (enter peripherally and radiate centrally). This can be done earlier in the introduction or in section 2.3.
  3. In section 2.1.3, the authors discuss nociceptive evoked potentials for both Ad and C fibers, but only then mention heat and electrical stimuli. It might be helpful to specify that electrical current can stimulate Ad neurons below pain thresholds and C fibers at and above pain thresholds at this point instead of waiting until 2.1.6.
  4. WST is not defined. 
  5. In addition to stating how many studies found significant differences in CNFL and CNFD, it would be very useful to state the number of subjects in the text as well. These numbers are in the table, but the strength of the outcomes should be disclosed in the main text. 
  6. The authors should mention that while IVCM detects the sub-basal plexus well, the intraepithelial terminals of the C fiber neurons after they leave the SBP cannot be imaged and thus, if a specific neuropathy/disease has a major effect on these terminals, it cannot be determined in this manner. 
  7. Do the authors think that some of the variability in the conclusions from multiple studies for PD and SS lies in different protocols/methodologies? Alternatively, could PD and SS have significant loss of the intraepithelial nerve terminals that cannot be detected by IVCM?

Author Response

Reviewer 3

Summary of Manuscript: The manuscript reviews the literature on the use of In Vivo Confocal Microscopy of corneal innervation for small fiber neuropathies. They compare the outcomes of various clinical testing procedures in detecting SFN, and recommend IVCM as a simpler, less invasive yet effective diagnostic tool. 

Critique Points:

  1. There are minor issues with the language throughout the text , and while it is easily comprehensible, it is noticeable. IF would be worthwhile to copy and language check the entire manuscript. A few examples are given here: 
    1. The authors use the term “resume: as in the short version of a curriculum vitae but they more likely mean summarize.  This should be corrected throughout the text.
    2. Check the use of sensibility throughout the manuscript, and correct to sensitivity.
    3. Line 176 “stimulate postganglionic sudomotor unmyelinated nerves C-fibers. 
    4. Line 269. A whorl-like pattern 1 about 102mm inferiorly…

Response to reviewer: We checked the manuscript and fixed the errors

  1. It might be helpful for the authors to give more detail on the autonomic fibers (peripheral and limbus) versus trigeminal fibers (enter peripherally and radiate centrally). This can be done earlier in the introduction or in section 2.3.

We thank the reviewer for this suggestion. We have added the brief detail on the autonomic innervation in section 2.3

  1. In section 2.1.3, the authors discuss nociceptive evoked potentials for both Ad and C fibers, but only then mention heat and electrical stimuli. It might be helpful to specify that electrical current can stimulate Ad neurons below pain thresholds and C fibers at and above pain thresholds at this point instead of waiting until 2.1.6.

Response to reviewer: This part was removed, based on the suggestion of another reviewer to focus the article on confocal microscopy and not on other methods

  1. WST is not defined. 

Response to reviewer: This part was removed based on the suggestion of another reviewer to focus the article on confocal microscopy and not on other methods

  1. In addition to stating how many studies found significant differences in CNFL and CNFD, it would be very useful to state the number of subjects in the text as well. These numbers are in the table, but the strength of the outcomes should be disclosed in the main text. 

Authors response:

The exact number of participants for each study is given in the tables. It is preferable to report the numerous data in tables for immediate evaluation for the reader. We think that repetition of so numerous data in the text would be confounding and would not add a piece of useful information for the readers being rather confounding. Additionally, it would stretch further the text that was already considered too long by another reviewer. However, if the reviewer believes that it would be useful for the manuscript, we will add this data in the text.

  1. The authors should mention that while IVCM detects the sub-basal plexus well, the intraepithelial terminals of the C fiber neurons after they leave the SBP cannot be imaged and thus, if a specific neuropathy/disease has a major effect on these terminals, it cannot be determined in this manner.

Thank you for this comment. We added this information in the chapter 2.4 and discussion.

  1. Do the authors think that some of the variability in the conclusions from multiple studies for PD and SS lies in different protocols/methodologies? Alternatively, could PD and SS have significant loss of the intraepithelial nerve terminals that cannot be detected by IVCM?

Reviewer 4 Report

The paper discusses the use of in vivo corneal confocal microscopy (IVCM) as a tool for investigating small-fiber neuropathy (SFN), a group of neurological disorders characterized by neuropathic pain symptoms and autonomic complaints. The authors of the paper argue that IVCM is an important tool for the diagnosis of SFN, as it allows for the immediate analysis of the quantity and morphology of corneal nerves, providing indirect information on the health of the peripheral nerves. The paper discusses the correlation between nerve density and morphology and the type of SFN, as well as the relationship between cataract and refractive surgery and iatrogenic dry eye disease. The authors conclude that IVCM is a valuable tool for the diagnosis and monitoring of SFN, and that it presents an opportunity for neurologists and other clinical specialists to improve their understanding and treatment of the condition.

pors:

The paper presents a valuable and novel approach to the diagnosis of small-fiber neuropathy, using in vivo corneal confocal microscopy to examine the quantity and morphology of corneal nerves. The authors of the paper provide a compelling argument for the usefulness of this method in investigating peripheral polyneuropathies, and their research presents an important step forward in the understanding and treatment of these conditions. Overall, the paper is a valuable contribution to the field and has the potential to improve the diagnosis and management of small-fiber neuropathy.

cons:

First, the research question and the significance of the study are not clearly explained in the introduction section. For example, authors should include how the use of in vivo corneal confocal microscopy for the diagnosis of small-fiber neuropathy differs from other diagnostic methods, how the use of IVCM for SFN diagnosis differs from other methods and why this is important with more details. Second, the relevant literature and the discussion about the recent studies are insufficient. For example, authors should discuss recent studies on the use of in vivo corneal confocal microscopy for the diagnosis of small-fiber neuropathy and its comparison to other diagnostic methods, recent studies on the use of IVCM for SFN diagnosis and its comparison to other methods, and recent progresses of using machine learning in the eye diseases studies[1][2]. Moreover, authors should describe the methods used in the study with more details, including the specific techniques used for corneal nerve analysis (e.g. IVCM), the criteria for defining SFN, and the follow-up measures used to monitor the progression of the condition. The experiments should include the correlation between nerve density and morphology and the type of SFN, disease duration, and follow-up. It is also suggested to discuss the implications of the results in more detail and their relevance to the field of neuropathy research, including the potential for the use of IVCM to improve the diagnosis and management of SFN.

[1] Li, Fei, et al. "A multicenter clinical study of the automated fundus screening algorithm." Translational Vision Science & Technology 11.7 (2022): 22-22.

[2] Han, Ruoan, et al. "Validating automated eye disease screening AI algorithm in community and in-hospital scenarios." Frontiers in Public Health 10 (2022).

Author Response

Reviewer 4

The paper discusses the use of in vivo corneal confocal microscopy (IVCM) as a tool for investigating small-fiber neuropathy (SFN), a group of neurological disorders characterized by neuropathic pain symptoms and autonomic complaints. The authors of the paper argue that IVCM is an important tool for the diagnosis of SFN, as it allows for the immediate analysis of the quantity and morphology of corneal nerves, providing indirect information on the health of the peripheral nerves. The paper discusses the correlation between nerve density and morphology and the type of SFN, as well as the relationship between cataract and refractive surgery and iatrogenic dry eye disease. The authors conclude that IVCM is a valuable tool for the diagnosis and monitoring of SFN, and that it presents an opportunity for neurologists and other clinical specialists to improve their understanding and treatment of the condition.

pors:

The paper presents a valuable and novel approach to the diagnosis of small-fiber neuropathy, using in vivo corneal confocal microscopy to examine the quantity and morphology of corneal nerves. The authors of the paper provide a compelling argument for the usefulness of this method in investigating peripheral polyneuropathies, and their research presents an important step forward in the understanding and treatment of these conditions. Overall, the paper is a valuable contribution to the field and has the potential to improve the diagnosis and management of small-fiber neuropathy.

 Thank for these comments supporting our paper.

cons:

First, the research question and the significance of the study are not clearly explained in the introduction section. For example, authors should include how the use of in vivo corneal confocal microscopy for the diagnosis of small-fiber neuropathy differs from other diagnostic methods, how the use of IVCM for SFN diagnosis differs from other methods and why this is important with more details.

Response to reviewer: The present paper is not a study but a review of actually available literature on the IVCM in SFN. The actually used diagnostic methods and IVCM characteristics were widely described in the introduction section. The overall methods description was shortened as suggested by other reviewers.

Second, the relevant literature and the discussion about the recent studies are insufficient. For example, authors should discuss recent studies on the use of in vivo corneal confocal microscopy for the diagnosis of small-fiber neuropathy and its comparison to other diagnostic methods, recent studies on the use of IVCM for SFN diagnosis and its comparison to other methods, and recent progresses of using machine learning in the eye diseases studies[1][2].

Response to reviewer: We have performed accurate literature research using both Pun Med and Medline database as stated at the chapter 2.5 Database and literature search.

We added the report of Lukashenko MV, Gavrilova NY, Bregovskaya AV et al. Corneal confocal microscopy in the diagnosis of small fiber neuropathy: faster, easier and more efficient than skin biopsy?. Pathophysiology. 2022;29:1-8.

We have also added the results of comparison of IVCM to other diagnostic methods used in SF diagnosis.

Additionally, we have added both suggested papers on machine learning in assessment of eye diseases as such approach is a very promising and quickly evolving diagnostic method that it will be undoubtedly of great interest in the next future also in the field of IVCM.

[1] Li F, Pan J, Yang D et al. "A multicenter clinical study of the automated fundus screening algorithm." Transl Vis Sci Technol.2022;11(7): 22.

[2] Han R, Cheng G, Zhang B et al. Validating automated eye disease screening AI algorithm in community and in-hospital scenarios.Front Public Health. 2022;10:944967

Moreover, authors should describe the methods used in the study with more details, including the specific techniques used for corneal nerve analysis (e.g. IVCM), the criteria for defining SFN, and the follow-up measures used to monitor the progression of the condition. The experiments should include the correlation between nerve density and morphology and the type of SFN, disease duration, and follow-up.

Response to the reviewer: This manuscript is a review of IVCM findings in small fibers neuropathies and not a research article. We did not perform experiments but examined all available articles reporting IVCM findings in different diseases that are characterized by SFN. So, these comments cannot be addressed.

It is also suggested to discuss the implications of the results in more detail and their relevance to the field of neuropathy research, including the potential for the use of IVCM to improve the diagnosis and management of SFN.

Response to the reviewer: We believe that the relevance to the field of neuropathy research is discussed sufficiently. We have highlighted the role of IVCM as an additional non-invasive tool recommended to diagnose the severity of small fibers damage and follow up of SFN.

Round 2

Reviewer 1 Report

Changes were made that clarify the findings

Reviewer 4 Report

Authors have well addressed my concerns in their revision. This version of paper presents a valuable and novel approach to the diagnosis of small-fiber neuropathy, using in vivo corneal confocal microscopy to examine the quantity and morphology of corneal nerves. The authors of the paper provide a compelling argument for the usefulness of this method in investigating peripheral polyneuropathies, and their research presents an important step forward in the understanding and treatment of these conditions. Overall, the paper is a valuable contribution to the field and has the potential to improve the diagnosis and management of small-fiber neuropathy.